# LinNet: Linear Network for Efficient Point Cloud Representation Learning

**Hao Deng**[1,2]    **Kunlei Jing**[3,4*]    **Shengmei Cheng**[1,2]    **Cheng Liu**[1,2]
**Jiawei Ru**[1,2]    **Jiang Bo**[1,2]    **Lin Wang**[1,2*]

[1]State-Province Joint Engineering and Research Center of Advanced Networking and Intelligent
Information Services, School of Information Science and Technology, Northwest University
[2]Shaanxi Key Laboratory of Higher Education Institution of
Generative Artificial Intelligence and Mixed Reality
[3]School of Software Engineering, Xi'an Jiaotong University
[4]Department of Computing, The Hong Kong Polytechnic University

denghao@stumail.nwu.edu.cn, kunlei.jing@xjtu.edu.cn
1615241805@qq.com, lc@nwu.edu.cn
rujiawei@stumail.nwu.edu.cn, {jiangbo, wanglin}@nwu.edu.cn

## Abstract

Point-based methods have made significant progress, but improving their scalability in large-scale 3D scenes is still a challenging problem. In this paper, we delve into the point-based method and develop a *simpler*, *faster*, *stronger* variant model, dubbed as **LinNet**. In particular, we first propose the disassembled set abstraction (DSA) module, which is more effective than the previous version of set abstraction. It achieves more efficient local aggregation by leveraging spatial anisotropy and channel anisotropy separately. Additionally, by mapping 3D point clouds onto 1D space-filling curves, we enable parallelization of downsampling and neighborhood queries on GPUs with linear complexity. LinNet, as a purely point-based method, outperforms most previous methods in both indoor and outdoor scenes without any extra attention, and sparse convolution but merely relying on a simple MLP. It achieves the mIoU of 73.7%, 81.4%, and 69.1% on the S3DIS Area5, NuScenes, and SemanticKITTI validation benchmarks, respectively, while speeding up almost 10x times over PointNeXt. Our work further reveals both the efficacy and efficiency potential of the vanilla point-based models in large-scale representation learning. *Our code will be available at* https://github.com/DengH293/LinNet.

## 1   Introduction

Appealed by the ongoing evolution progress of technologies in robotics, autonomous driving, augmented reality, etc., LiDAR sensors are incrementally integrated into hardware-constrained devices such as mobile devices and AR headsets. This has led to a growing interest in efficient point cloud processing models. Given the limited computational power of mobile devices and embedded systems, the design of mobile-friendly point cloud representation learning algorithms should not only focus on performance but also pay attention to high computational efficiency.

Unlike images, point cloud data is irregular and unordered. There are various methods for processing point cloud data in 3D vision. Common approaches include multi-view methods [1, 2, 3] and voxel-based methods [4, 5]. Converting irregular data into the required formal representations often

---

*Corresponding Authors

38th Conference on Neural Information Processing Systems (NeurIPS 2024).

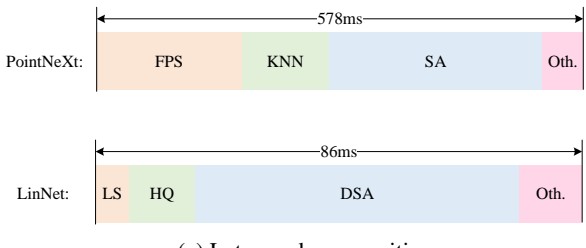
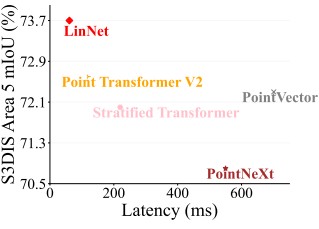

(a) Latency decomposition.                             (b) Accuracy-speed tradeoff.

Figure 1: **Latency decomposition.** We show the inference run time decomposition. (a) SA: set abstraction; DSA: disassembled set abstraction; LS: linearization sampling; HQ: hash query; Oth.: others. (b) LinNet achieves the highest mIoU with extremely low latency compared to the comparative point-based approaches. The latency of each network is measured on a single Nvidia 3090 GPU, taking a batch of 80k points.

requires additional computation and memory and thus results in the loss of geometric information [6]. Therefore, point-based methods that directly operate on point clouds have emerged. PointNet [7] and PointNet++ [8] are the pioneers of this approach, introducing a general point cloud learning paradigm from local to global. The paradigm consists of two parts: the first part is a spatial neighborhood search strategy, which utilizes algorithms such as furthest point sampling (FPS) and K-nearest neighbors (KNN) to implement sub-sampling and neighborhood grouping of point clouds, respectively. The second part is a trainable local feature extractor. Following them, subsequent extensive research [9, 10, 11, 12, 13, 14, 15, 16, 17, 18, 19] have shown promising results by focusing on the design of more sophisticated extractors.

Despite the impressive results that have been achieved in object recognition and semantic segmentation, most of these methods are limited to applications in small-scale 3D point clouds. The main reason is discouraged by the high time complexity of the neighborhood search strategy that the point-based methods adopted. As shown in Fig. 1a. FPS and KNN occupy 46% of the runtime. This draws forth the main motivation of this paper: *enhancing the scalability of point-based approaches in large-scale scenes, while maintaining their excellent performance in small-scale tasks*.

In this paper, we introduce a novel model, named ***Linear Net (LinNet)***. Our LinNet derives from inheriting the innovations and overcoming the drawbacks of the PointNet++ paradigm, including a disassembled set abstraction (DSA) module and an efficient point search strategy. Specifically, inspired by MobileNet [20], we first use two independent MLPs to separately learn depth-wise geometric features of the neighborhood and point-wise semantic features. Then, the geometric features are assigned as biases to the queried neighbors' semantic features, achieving spatially anisotropic neighborhood aggregation. Since the learning of high-dimensional semantic features is point-wise and does not involve the neighborhood, the required floating-point operations (FLOPs) are significantly lower than those needed for SA. Besides, a hash query and linearization sampling strategy are proposed for speeding up point searching. The core of our method is to map the 3D search space onto a segmented curve for acceleration. A sparse point cloud is ordered on that curve, and points adjacent to each other in the curve are also adjacent in space. For neighborhood queries, we store each segmented curve as a bucket in a hash table. When querying the neighborhood, we only need to search in the buckets corresponding to the neighboring curves, which drastically cuts down the search range. Linear sampling ensures uniform sampling by taking the point closest to the origin within each grid as the new sampling point. The method reduces the time complexity to be linear and supports GPU parallelism, resulting in very fast sampling. As shown in Fig. 1, the additional point cloud search operations take up less than 10% of the model's runtime. By employing these techniques, our method achieves efficient point cloud representation learning and scalability, providing significant performance improvements for large-scale point cloud analysis.

The contribution of our paper can be summarized in the following three folds:

- We analyze the feature aggregation of the vanilla SA module and introduce a novel efficient and effective DSA module. This strategy effectively reduces computational overhead and achieves performance gains. Moreover, we discuss the superiority of this method from the perspective of weight initialization, emphasizing how these adjustments crucially enhance the overall performance of the network.

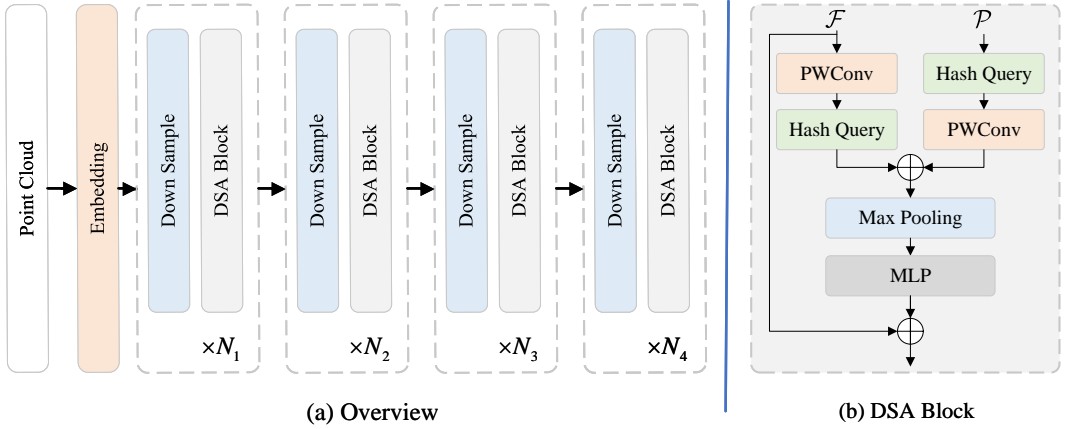

Figure 2: **Overall architecture.** (a) Overview of the framework. The whole network consists of an embedding layer and four stages, each containing a downsampling layer and $N_i$ disassembled SA blocks. (b) Structure of the DSA blocks. Each DSA block consists of a DSA module and extra MLPs.

- To improve scalability in large-scale scenes, a linear complexity point cloud search strategy is introduced, mapping a 3D point cloud to a 1D space-filling curve. This approach drastically reduces the time consumption associated with sub-sampling and neighbors query.

- Experiments show that our approach achieves state-of-the-art performance on widely adopted 3D large-scale semantic segmentation benchmarks (S3DIS, NuScenes) and competitive results on small-scale classification tasks (ScanObjectNN and ModelNet40). Extensive ablation studies have also validated the effectiveness of our proposed components.

## 2    Related Works

**Point cloud analysis.** Point cloud analysis is primarily approached in two ways. Another approach, exemplified by the PointNet family, directly processes raw point clouds. They introduce a hierarchical feature learning paradigm to recursively capture local geometric structures. By adopting local point representation and multi-scale information, PointNet++ has demonstrated excellent performance and has become a cornerstone of modern point cloud methods [9, 16, 21, 17, 22, 23]. Our LinNet follows the design philosophy of PointNet++ but explores a simpler yet deeper network architecture.

**Voxel-based methods.** Instead of learning directly on discrete points, the sparse convolution [5, 24] first converts the point cloud into a regular grid and then constructs a full convolutional neural network using the discrete sparse tensor. By building a hash table of discrete rasters, neighborhood query and sampling can be performed efficiently with a constant time complexity of $\mathcal{O}(1)$. In addition, the hash table construction and query can be implemented in parallel on CUDA, which significantly improves the computational efficiency. However, even though sparse convolution performs well in many large-scale point cloud tasks, it still faces challenges in capturing the fine-grained patterns of point clouds. This is due to the quantization artifacts that may be introduced during voxelization, resulting in the extracted features being limited by the voxel size [25].

**Efficient network in computer vision.** In computer vision, an efficient network typically refers to a deep learning architecture designed to balance performance and operational efficiency, including aspects like latency, FLOPs, memory, and power consumption. MobileNet [20] use depthwise separable convolutions, making them particularly efficient for mobile and embedded devices. EfficientNet [26] systematically scales the network's width, depth, and resolution, achieving a balance between efficiency and accuracy. Many works in 3D vision [27, 18, 19, 25] are dedicated to optimizing the efficiency and performance of point cloud processing. While such networks often slightly compromise on performance for reduced computational load, our network, as detailed in Section 4, uniquely enhances both efficiency and accuracy.

# 3 Linear Net

## 3.1 Problem Formulation

Given a 3D point cloud $\mathcal{V} = (\mathcal{P}, \mathcal{F})$ consisting of $n$ points $\boldsymbol{x}$. For the $\boldsymbol{i}$-th point $\boldsymbol{x}_i = (\boldsymbol{p}_i, \boldsymbol{f}_i)$, $\boldsymbol{p}_i \in \mathbb{R}^3$ and $\boldsymbol{f}_i \in \mathbb{R}^c$ are the space coordinates and features, respectively. The task of point cloud semantic segmentation involves assigning a class label to each point $\boldsymbol{x}_i$, while scene classification entails predicting a class label for the entire scene $\mathcal{C}$. The point-based methods usually employ several stages to classify the points or point clouds. In each stage, a downsample layer is first applied to sample the points, reducing the density of the point cloud.

## 3.2 Disassembled Set Abstraction

In this section, we introduce the DSA modules. First, we progressively explore the direct application of separable convolutions in 3D vision. Then, we adopt a harmonized approach to separate channel and spatial anisotropy. Finally, we discuss the superiority of this method in terms of weight initialization, showing why the adjustment can improve overall network performance.

**Revisiting Local Aggregation of Computer Vision.** In a standard convolutional kernel, the anisotropy of the weights plays a critical role in capturing local information. This anisotropy can be classified into two categories: spatial and channel anisotropy. Spatial anisotropy refers to the variations among the features of neighboring points within the same feature channel, while channel anisotropy reflects the differences across various feature channels. To improve efficiency, MobileNet [20] achieves speedup by decomposing the standard convolutional kernel: it divides the convolutional kernel into point-wise convolution (PWConv) for channel anisotropy and depth-wise convolution (DWConv) for spatial anisotropy. Due to the sparsity of point clouds, it is impractical to apply DWConv to handle them directly. Instead of achieving anisotropy through parameters, point-based methods approach this by manipulating the input directly and adding anisotropy to the input data. Given a point cloud $\boldsymbol{x}_i$, a typical local aggregation in 3D vision can be formulated as:

$$\boldsymbol{f}'_i = \mathcal{R}_{j:(i,j)\in\mathcal{N}}\{\text{PWConv}^{3+c\mapsto c}(\boldsymbol{f}_j||(\boldsymbol{p}_j - \boldsymbol{p}_i))\}, \tag{1}$$

where $\mathcal{R}$ is max-pooling that aggregates the local feature from the neighbors of anchor point $\boldsymbol{x}_i$ denoted as $\{j : (i,j) \in \mathcal{N}\}$. $||$ is the concatenate operation in channels. $\text{PWConv}^{3+c\mapsto c} : \mathbb{R}^{3+c} \mapsto \mathbb{R}^c$ is an MLP that consists of pointwise convolution, batch normalization layer, and ReLU activation function. Here, the neighborhood features of the different anchors come from the same query set, so they are isotropic. The coordinates are anisotropic as they are relative to their respective anchor. Concatenating together the isotropic features and anisotropic coordinates gives the input anisotropy.

**Depth-wise Separate Set Abstraction.** Corresponding to the separate weights, we initially chose to separate the inputs directly as follows:

$$\boldsymbol{f}'_i = \mathcal{R}_{j:(i,j)\in\mathcal{N}}\{\text{PWConv}^{c\mapsto c}(\boldsymbol{f}_j)\} + \mathcal{R}_{j:(i,j)\in\mathcal{N}}\{\text{PWConv}^{3\mapsto c}((\boldsymbol{p}_j - \boldsymbol{p}_i))\}. \tag{2}$$

Since all $f'_j$ come from the same set, Eq. (2) is identity to Eq. (3):

$$\begin{aligned} \overline{\boldsymbol{f}}_i &= \text{PWConv}^{c\mapsto c}(\boldsymbol{f}_i); \\ \boldsymbol{f}'_i &= \mathcal{R}_{j:(i,j)\in\mathcal{N}}\{\overline{\boldsymbol{f}}_j\} + \mathcal{R}_{j:(i,j)\in\mathcal{N}}\{\text{PWConv}^{3\mapsto c}((\boldsymbol{p}_j - \boldsymbol{p}_i))\}. \end{aligned} \tag{3}$$

This separation is necessary for several reasons. First, since the features are derived from the same query set, there is no need to apply a shared-weight PWConv on the neighboring features [18]. If the feature dimension is $c$ and the number of neighbors is $k$, the computational complexity would be $kc^2$. After separation, this complexity is reduced to $c^2$. Second, the distribution patterns of coordinates and features differ significantly, and using independent convolutional kernels allows for better capture of these differences. However, as encountered with MobileNet, the speedup often comes at the cost of

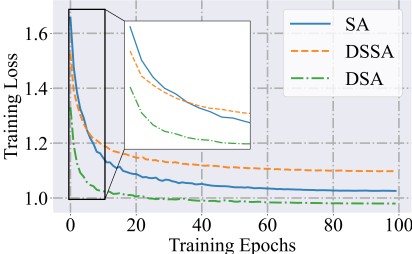

Figure 3: **Training on S3DIS**.

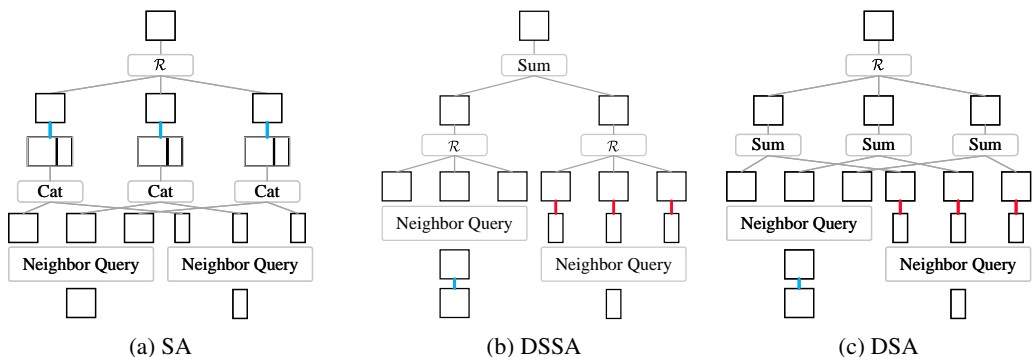

| (a) SA | (b) DSSA | (c) DSA |

Figure 4: **Comparison of various local aggregation.** Each square represents the semantic feature, while each rectangle represents relative coordinates. The number of neighbors is 3. The blue line indicates a mapping in a high-dimensional space (e.g., from $3 + c$ to $c$, or $c$ to $c$), and the red line indicates a mapping from a lower dimension to a higher dimension (e.g., from 3 to $c$). More blue lines indicate more computation.

reduced accuracy. As illustrated in Fig. 3, DSSA accelerates the model's convergence during the initial few epochs. Yet, as training progresses, the convergence slows down, and the model's overall performance starts to degrade. The main reason lies in *the lack of spatial-wise anisotropy between neighbor features*. When features are processed independently of their spatial relationships, it can lead to a loss of contextual information critical for certain tasks, such as those involving complex spatial structures or detailed textural information.

**Disassembled set abstraction.** Building on the principles outlined above, we propose a novel method for lightweight local aggregation that addresses the inherent challenges of separating coordinates and features. The proposed disassembled set abstraction (DSA) module can be formulated as:

$$\overline{\boldsymbol{f}}_i = \text{PWConv}^{c \mapsto c}(\boldsymbol{f}_i);$$
$$\boldsymbol{f}'_i = \text{BN}\{\mathcal{R}_{j:(i,j) \in \mathcal{N}}\{\overline{\boldsymbol{f}}_j + \text{PWConv}^{3 \mapsto c}((\boldsymbol{p}_j - \boldsymbol{p}_i))\}\}, \tag{4}$$

where BN is a batch normalization layer [28]. The spatial anisotropy derived from relative positions is integrated into the neighborhood feature aggregation as a manner of bias. This integration ensures that the aggregation of neighborhood features is closely linked to the spatial distribution of the point cloud, thereby enhancing the model's robustness under varying spatial distributions. Interestingly, the Eq. (1) and Eq. (4) are actually mathematically equivalent during forward propagation. However, the DSA module exhibits faster convergence and lower loss compared to the SA modules (see Fig. 3). This phenomenon can be attributed to **variations in the initialization of weights** as follows.

Excluding bias, Eq. (1) uses a combined weight matrix $\mathbf{W}$ (dimensions $c \times (c+3)$) to process semantic and geometric data simultaneously. For a given neighbor $j$, the input vector $\mathbf{x}_j = [\boldsymbol{f}_j, \boldsymbol{p}_j - \boldsymbol{p}_i]$ results in the output:

$$\mathbf{y}_j = \mathbf{W}\mathbf{x}_j^{\text{T}}. \tag{5}$$

In this case, Kaiming initialization sets $\mathbf{W}$ as a normal distribution $\mathcal{N}(0, \sqrt{\frac{2}{c+3}})$, potentially reducing the impact of geometric data due to its smaller proportional weight. In stark contrast, the DSA module separates the processing of semantic and geometric data using two distinct weight matrices, $\mathbf{W}_f$ for semantic (dimensions $c \times c$) and $\mathbf{W}_p$ for geometric data (dimensions $c \times 3$), leading to:

$$\mathbf{y}_j = \mathbf{W}_f \boldsymbol{f}_j^{\text{T}} + \mathbf{W}_p (\boldsymbol{p}_j - \boldsymbol{p}_i)^{\text{T}}. \tag{6}$$

The specific initialization $\mathbf{W}_f \sim \mathcal{N}(0, \sqrt{\frac{2}{c}})$ and $\mathbf{W}_p \sim \mathcal{N}(0, \sqrt{\frac{2}{3}})$ allows for a more balanced influence of geometric data, thus enhancing the network's ability to extract and utilize geometric information effectively. It is noteworthy that the PWConv between high-dimensional semantic features is applied directly to the point cloud features, rather than to the neighborhood. Additionally, the number of input channels for PWConv applied to the neighborhood is only 3, which is significantly smaller than $c$ (with a minimum of 64 in the segmentation model). As illustrated in Fig 4, DSA requires substantially fewer FLOPs compared to SA, thereby improving computational efficiency and making it more suitable for large-scale applications.

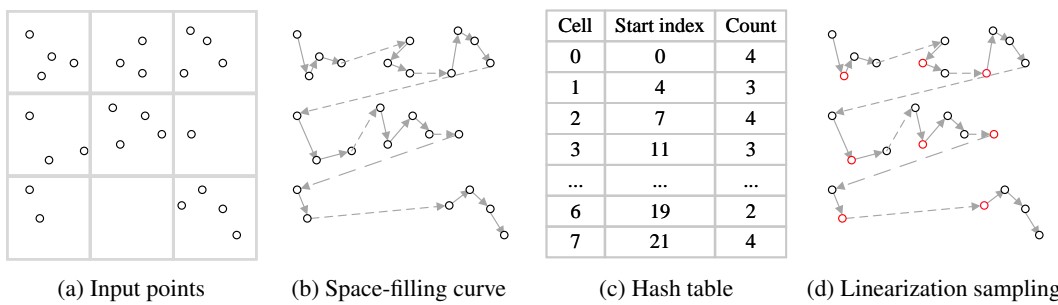

| Cell | Start index | Count |
|------|-------------|-------|
| 0 | 0 | 4 |
| 1 | 4 | 3 |
| 2 | 7 | 4 |
| 3 | 11 | 3 |
| ... | ... | ... |
| 6 | 19 | 2 |
| 7 | 21 | 4 |

(a) Input points     (b) Space-filling curve     (c) Hash table     (d) Linearization sampling

Figure 5: **Efficient point clouds searching for query and sampling.** (a) The input point cloud. (b) Point cloud after linearization by space-filling curves. Points connected by solid arrows are within the same grid, while dashed lines connect points between different grids. (c) Store each segment of the solid line as a hash table. (d) The point closest to the center of each region represented by segments connected by solid arrows is chosen as the new sampling point.

## 3.3 Point Searching Strategy

Recent literature [29, 30] employ space-filling curves to serialize point clouds, which are then uniformly divided into patches and fed into a transformer architecture. Inspired by this, we map points in 3D space onto a space-filling curve for accelerating point searching. Denote the coordinates $\boldsymbol{p}$ as $(x, y, z)$ and the batch index as $b$. The shuffled key is defined as a 64-bit integer:

$$\text{Key} = (b \ll 54) \,|\, (\lfloor z/s \rfloor \ll 36) \,|\, (\lfloor y/s \rfloor \ll 18) \,|\, (\lfloor x/s \rfloor), \tag{7}$$

where $\ll$ denotes left bit-shift, $s$ denotes the grid size, and $|$ denotes bitwise OR. It is worth noting that, unlike PTv3 [30], which contains only one point per grid cell, our approach allows multiple points to share the same key. We utilize shuffled keys to store these points in memory in an ordered manner, ensuring that elements sharing the same key are close to each other in memory. As shown in Fig. 5b, points on the same solid line are in the same grid. Through this, the point cloud $\mathcal{V}$ is partitioned into $m$ sub-regions $[\mathcal{V}_1, \mathcal{V}_2, \ldots \mathcal{V}_m]$.

**Hash query.** By storing coordinates in spatial order, a hash table can be constructed to efficiently manage and query neighbors. Each bucket represents a non-empty grid. The key is the shuffled key of the grid, and the value contains two parts: the index of the first point in the grid and the count of points in that grid. If the number of grids is $m$, the time complexity of building the hash table is $\mathcal{O}(m)$. During the query phase, for each point, we find the 27 (i.e., $3 \times 3 \times 3$) neighboring grids and query the hash table with these keys. Finally, the top $k$ nearest points are selected. Assuming each point's 27-grid neighborhood contains $p$ points on average, identifying the closest $k$ points involves maintaining a heap with a complexity of $\mathcal{O}(p \log k)$ and a final sorting step costing $\mathcal{O}(k \log k)$. Thus, the total computational complexity is $\mathcal{O}(m + N(p \log k + k \log k))$, while that of $k$NN is $\mathcal{O}(kN^2)$.

**Linearization sampling.** To achieve uniform and fast sampling, we select a point from each subset according to the following rule:

$$\text{idx}_i = \arg \min_{j \,:\, (j) \in \mathcal{M}_i} \left( \|\boldsymbol{p}_j - \lfloor \boldsymbol{p}_j/s \rfloor \times s\|_2 \right), \tag{8}$$

where $(j) \in \mathcal{M}_i$ are the points of $i$-th sub-regions. This rule ensures that the newly sampled points are the closest to the origin within their respective grids, guaranteeing uniform sampling. The method has a linear time complexity and supports GPU parallelism, resulting in very fast sampling speeds.

## 3.4 Network Architecture

The overall architecture is illustrated in Fig. 2. For segmentation tasks, we use both encoders and decoders. To ensure a fair comparison, in the indoor dataset S3DIS, we configure the encoder depth as [4, 7, 4, 4], which is the same as PointNeXt. For the outdoor dataset, the encoder depth is set to [4, 4, 7, 4]. Specifically, the channel numbers for these stages are set to [C, 2C, 4C, 8C], with C being 64. For the classification task, only the encoder is used. Considering that the dataset for the classification task is small and PointNeXt is already capable of real-time response, we do not use the proposed linear search strategy, ensuring a fairer comparison between the DSA module and the SA module.

Table 1: Indoor sem. seg. on S3DIS Area 5.

| Methods | Input | mIoU | OA | Acc |
|---------|-------|------|-----|-----|
| PointNet [7] | point | 41.1 | - | 66.2 |
| PointCNN [31] | point | - | - | 75.6 |
| PointWeb [32] | point | 60.3 | 87.0 | 66.6 |
| PointNet++ [8] | point | 68.6 | 87.7 | 67.1 |
| KPConv [33] | point | 67.1 | - | 79.1 |
| RandLA-Net [27] | point | - | - | 82.0 |
| PTv1 [9] | point | 70.4 | 90.8 | 76.5 |
| CBL [34] | point | 69.4 | 90.6 | - |
| PointMeta [17] | point | 72.0 | 91.4 | - |
| ASSANet [18] | point | 66.8 | - | - |
| Str. Trans. [21] | point | 72.0 | 91.5 | 78.1 |
| Fast PT [25] | point | 70.1 | - | 77.3 |
| PTv2[‡] [11] | point | 72.6 | 91.6 | 78.0 |
| PTv3[‡] [30] | point | 73.4 | - | - |
| ConDaFormer[‡] [35] | point | 73.5 | **92.4** | 78.9 |
| PointVector [16] | point | 72.3 | 91.0 | 78.1 |
| PointNeXt [22] | point | 70.8 | 91.7 | 77.5 |
| LinNet (ours) | point | 72.9 | 91.3 | 78.6 |
| LinNet [‡] (ours) | point | **73.7** | 91.9 | **79.0** |

Table 2: Outdoor sem. seg. on NuScenes.

| Methods | Input | Val | Test |
|---------|-------|-----|------|
| RandLA-Net [27] | point | - | - |
| KPConv [33] | point | - | - |
| RangeNet++ [36] | point | 65.5 | - |
| Salsanext [37] | hybrid | 72.2 | - |
| MinkUNet [4] | voxel | 73.3 | - |
| PolarNet [38] | point | 71.0 | - |
| PVKD [39] | hybrid | 76.0 | - |
| AMVNet [40] | - | - | 76.1 |
| Cylender3D [41] | cylender | 76.1 | 77.2 |
| SPVNAS [42] | hybrid | 77.4 | - |
| RPVNet [43] | hybrid | 77.6 | - |
| 2DPASS[‡] [44] | hybrid | - | 80.8 |
| RangeFormer [45] | - | 78.1 | 80.1 |
| SphereFormer [46] | voxel | 78.4 | 81.9 |
| WaffleIron[‡] [47] | point | 79.1 | - |
| OACNN[‡] [48] | voxel | 78.9 | - |
| PTv3[‡] [30] | point | 80.4 | **82.7** |
| LinNet(ours) | point | 80.4 | - |
| LinNet [‡](ours) | point | **81.4** | 82.3 |

## 4 Experiments

To validate the effectiveness of LinNet, we conduct experiments in 3D semantic segmentation and 3D object classification tasks. We also conduct an extensive ablation study to analyze each component in LinNet. More details of the experiments can be found in the Appendix.

### 4.1 Semantic Segmentation

**Data and metric.** S3DIS [49] (Stanford Large-Scale 3D Indoor Spaces) is a challenging benchmark that comprises 6 extensive indoor areas, 271 rooms, and 13 semantic categories, which represent different types of objects and room elements commonly found in indoor environments. Each point in the dataset is labeled with one of the 13 semantic categories, such as table, door, chair, column, and window, in addition to clutter. Following previous work [22], we subsample the grid before sending the point cloud to the network. The grid size and maximum number of points are set to 0.04m and 24000 respectively. During training, we crop the center point by the pre-set maximum number of points and discard the rest. During testing, the entire scene is processed. The experiment results are shown in Tab. 1. For evaluation metrics, we choose class-wise intersection over union (mIoU), mean of class-wise accuracy (mAcc), and overall point-wise accuracy (OA). Given that S3DIS is relatively small, we conduct further experiments on the NuScenes [50] dataset to validate the efficiency of our model. In this dataset, each point is annotated with one of the 16 semantic categories. The dataset encompasses 1,000 scenes collected in Boston and Singapore, reflecting diverse urban environments. We adhered to the official segmentation protocol, allocating 700 scenes for training, 150 for validation, and another 150 for testing, ensuring a balanced and comprehensive evaluation of our model's performance across varied scenes.

**Performance.** The results are shown in Tab. 1 and Tab. 2. Following Point Transformer v2 [11] and CondaFormer [35], we also employ the test time augmentation (TTA) strategy to achieve fairer comparisons, and results using the TTA strategy are labeled with [‡]. In indoor dataset S3DIS, our LinNet outperforms the SoTA point-based method PointNeXt [22] by 2.9%, 0.2%, 2.5% in terms of mIoU, OA, and mAcc, respectively. We also visualize the segment result in Fig. 6. In large-scale dataset NuScenes, LinNet also outperforms all previous methods. It is worth noting that our approach utilizes a pure MLP network, employing solely point-wise convolutions. This highlights that sophisticated feature extractors, such as attention mechanisms and graph structures, are not essential for achieving robust segmentation capabilities. Moreover, our method is strictly based on point data, devoid of any sparse convolutions, which further underscores the scalability of point-based methods in managing large-scale point clouds effectively.

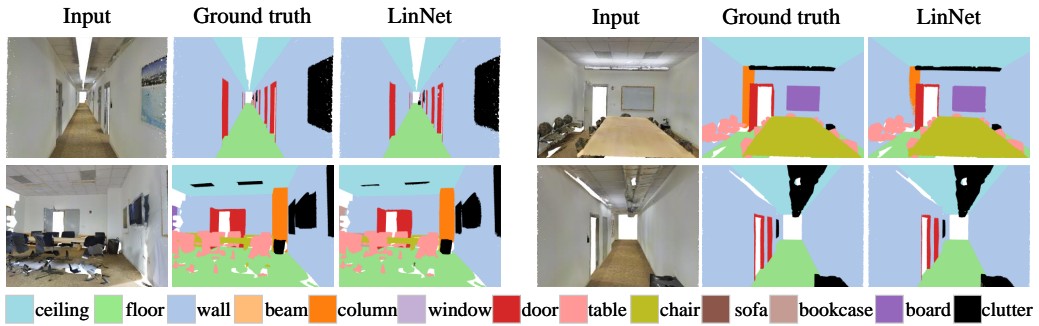

| | Input | Ground truth | LinNet | Input | Ground truth | LinNet |

ceiling floor wall beam column window door table chair sofa bookcase board clutter

Figure 6: Comparative Visualization of Semantic Segmentation on S3DIS.

Table 3: **3D object classification in ScanObjectNN and ModelNet40.** Averaged results in three random runs using 1024 points as input without normals and without voting are reported.

| Method | ScanObjectNN (PB_T50_RS) | | ModelNet40 | | Params. M | FLOPs G | Throughput (ins./sec.) |
| | OA (%) | mAcc (%) | OA (%) | mAcc (%) | | | |
|---|---|---|---|---|---|---|---|
| PointNet [7] | 68.2 | 63.4 | 89.2 | 86.2 | 3.5 | 0.9 | 4199 |
| PointCNN [14] | 78.5 | 75.1 | 92.2 | 88.1 | 0.6 | - | 44 |
| DGCNN [13] | 78.1 | 73.6 | 92.9 | 90.2 | 1.8 | 4.8 | 458 |
| DeepGCN [51] | - | - | 93.6 | 90.9 | 2.2 | 3.9 | - |
| KPConv [33] | - | - | 92.9 | - | 14.3 | - | - |
| ASSANet-L [18] | - | - | 92.9 | - | 118.4 | - | 144 |
| SimpleView [3] | 80.5±0.3 | - | 93.0±0.4 | 90.5±0.8 | 0.8 | - | - |
| MVTN [1] | 82.8 | - | 93.5 | 92.2 | 3.5 | 1.8 | 2- |
| Point Cloud Transformer [10] | - | - | 93.2 | - | 2.9 | 2.3 | - |
| CurveNet [52] | - | - | 93.8 | - | 2.0 | - | - |
| PointMLP [23] | 85.4±1.3 | 83.9±1.5 | **94.1** | 91.3 | 13.2 | 31.3 | 220 |
| PointMetaBase [17] | 87.9±0.2 | - | - | - | - | 0.6 | - |
| PointNeXt [22] | 87.7±0.4 | 85.8±0.6 | 93.7±0.3 | 90.9±0.5 | 1.4 | 1.6 | 2126 |
| LinNet (ours) | **88.2±0.4** | **86.6±0.7** | 93.6±0.2 | **91.0±0.5** | 1.4 | 0.6 | 1852 |

## 4.2 Object classification

We chose the ScanObjectNN [53] and ModelNet40 [54] datasets to assess the classification capabilities of our model, and the results are shown in Fig. 3. We also report the parameters, FlOPs, and throughput. Following PointNeXt, the input channels of the models used in ScanobjectNN and ModelNet40 are 32 and 64 respectively. The model parameters are computed for C = 32.

**ScanObjectNN.** It contains approximately 15,000 real scanned objects divided into 15 categories with 2,902 instances, which presents substantial challenges due to occlusions and noise. We conduct experiments on PB_T50_RS, the most challenging and frequently used variant of ScanObjectNN. According to the report, the proposed LinNet significantly outperformed existing methods in Overall Accuracy (OA) and mean Accuracy (mAcc), using fewer model parameters and achieving faster processing speeds. LinNet achieved an OA of 88.6% and a mAcc of 87.3% on ScanObjectNN. Note that we employ the same training protocols and experimental conditions as the SOTA benchmark, PointNeXt. Nonetheless, we still achieve an OA improvement of 0.4%, while other PointNeXt style architectures (e.g., PointMetaBase [17]) using the same experimental setup only achieve an OA improvement of 0.1%.

**ModelNet40.** This dataset is a widely-used object classification dataset, comprises 12,311 3D computer graphics CAD models across 40 categories. Our results, as indicated, are highly competitive and exceed those of most previous methods. LinNet achieved an Overall Accuracy of 93.9% on ModelNet40, surpassing graph-based models like DGCNN [13], transformer-based models such as Point Transformer [9], and KPConv [33].

## 4.3 Model Efficiency

We evaluate the efficiency of our model at four different scales of points: 20k, 50k, 100k, and 200k. The down sampling rate is about 4. The experiments are performed on an RTX 3090. The models we compared include PointNeXt-XL [22] and Point Transformer v2 [11]. As shown in Fig. 7a, the

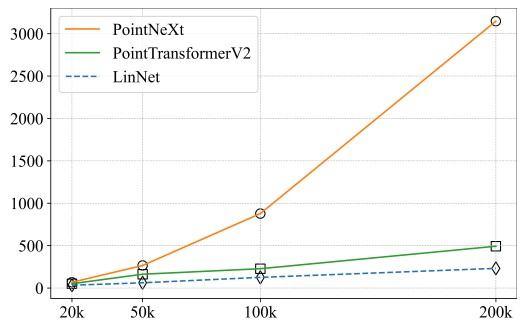
(a) Efficiency comparison of different models.

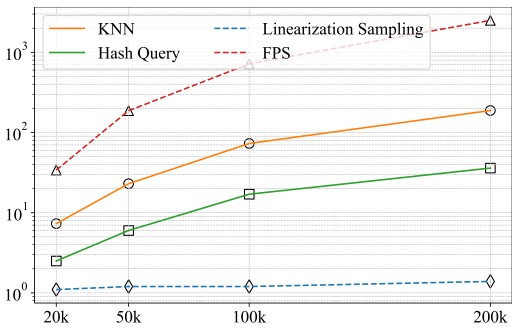
(b) Efficiency comparison of different components.

Figure 7: **Efficiency comparisons.** The horizontal axis represents the number of points in the input tensor and the vertical axis represents the running time in milliseconds.

Table 4: Model design ablation.

| ID | LS | HQ | DDSA | DSA | mIoU | Latency(ms) |
|---|---|---|---|---|---|---|
| I | | | | | 70.8 | 89 |
| II | ✓ | | | | 72.0 | 45 |
| III | | ✓ | | | 70.8 | 80 |
| IV | ✓ | ✓ | | | 72.0 | 38 |
| V | ✓ | ✓ | ✓ | | 71.0 | 32 |
| VI | ✓ | ✓ | | ✓ | 73.1 | 34 |

Table 5: Ablation on the DSA design.

| ID | Methods | mIoU | Δ |
|---|---|---|---|
| (1) | Vanilla SA | 72.0 | -1.1 |
| (2) | DSSA | 71.0 | -2.1 |
| (3) | ASSA | 70.5 | -2.6 |
| (4) | PosPool | 69.9 | -3.2 |
| (5) | Avg. pooling | 71.2 | -1.9 |
| (6) | DSA | 73.1 | - |

latency of LinNet grows linearly with the scale of the point cloud. Point Transformer v2 employs grid pooling, a technique similar to our linearization sampling, both characterized by linear time complexity. However, thanks to the simplicity and efficiency of our disassembled set aggregation, our model exhibits only half the latency of Point Transformer v2. Notably, at the 200k level, our LinNet model operates 13 times faster than PointNeXt, demonstrating significant improvements in processing speed. We also explore the latency of the proposed point cloud search strategy. Note that the horizontal axis is on a logarithmic scale. As shown in Fig. 7b, linearization sampling and hash query based on space-filling curves leads to greater speedup as the point cloud size increases. The proposed linearization sampling can be up to a thousand times faster than FPS.

## 4.4 Ablation Study

We conduct ablation experiments of the model to verify the validity of each component, and all experimental results are averaged over three times in the S3DIS area 5 unless otherwise stated.

**LinNet.** We perform ablation experiments on different modules introduced in LinNet: linearization sampling (LS), hash query (HQ), depth-wise separate set abstraction (DSSA), and disassembled set abstraction (DSA), with the results shown in Tab. 4. The delay was measured on a 20k number of point clouds. The model used in Exp. I is PointNeXt, which is the baseline result of our design. In Exp. II through VI, we progressively incorporated each of our proposed components, improving the baseline accuracy to 73.1% and reducing the latency to 34 ms.

**Disassembled set abstraction.** In Tab. 5, we investigate the design of the feature aggregation module to improve the aggregation of semantic and geometric information. We utilize ASSA [18] and PosPool [19], instead of DSA module. Additionally, we evaluate the performance impact of replacing max pooling with average pooling. Exp. (1) and (2) show a 1% decrease in accuracy when using simple separation of inputs directly. Comparing Exp. (1) with Exp. (6) indicates that the proposed DSA module performs better than the SA module. Exp. (3) and (4), which employ ASSA and PosPool respectively, demonstrate performance degradation, highlighting that our data-driven approach outperforms the parameter-free strategy in merging low-dimensional coordinates and high-dimensional semantics. Exp. (5) shows that max pooling is more compatible with our network.

**Model scalability.** We refer to the default LinNet as LinNet-Base and designed two variants with different numbers of trainable parameters: LinNet-Small, which has one-tenth the parameters of LinNet-Base, and LinNet-Large, which has four times the parameters of LinNet-Base. We test the

Table 6: Model scalability. Latency and FLOPs are measured with 24k points.

| Name | Channels | Depths | Param(M). | FLOPs (G) | Latency (ms) | mIoU(%) |
|------|----------|--------|-----------|-----------|--------------|---------|
| Small | 32 | [2,2,2,2] | 1.5 | 2.1 | 27 | 77.6 |
| Base | 64 | [4,4,7,4] | 16.5 | 7.8 | 34 | 80.4 |
| Large | 128 | [4,4,7,4] | 65.6 | 24.9 | 42 | 81.3 |

Table 7: Memory footprint during training and inference on the NuScenes dataset.

| Model | Training Memory | Inference Memory |
|-------|-----------------|------------------|
| MinkUNet | 2.6 GB | 1.4 GB |
| PointNeXt | Out of Memory | Out of Memory |
| LinNet-Small | 5.2 GB | 4.9 GB |
| LinNet | 16 GB | 13 GB |

performance of these models on the outdoor dataset NuScenes, and the results are summarized in Tab. 6. The mIoU on the validation set steadily improves with increasing model size. Additionally, since the models are linear, the increase in parameters does not result in significantly higher latency. Notably, without using TTA, our LinNet-Small achieves a validation accuracy of 77.6% with only 1.7M parameters, surpassing the 38M parameter sparse convolution method MinkUnet [4].

**Memory footprint.**    We conduct experiments to evaluate memory usage during both training and inference phases on the NuScenes dataset, utilizing an RTX 4090 graphics card with all tests conducted at a batch size of 1. We include comparisons with the baseline model PointNeXt [22] and the sparse convolution method MinkUNet [4]. Our findings reveal that PointNeXt suffers from out-of-memory issues when handling large-scale scenes, highlighting scalability challenges. In contrast, our DSA module significantly reduces memory consumption by avoiding high-dimensional feature transformations on neighboring point clouds. Given that MinkUNet starts with 32 input channels, we conducted similar tests with our LinNet-Small model, which also has 32 initial feature channels, for a direct comparison. The results are shown in Tab. 7. Although LinNet-Small consumes more memory than MinkUNet, it is crucial to note that LinNet-Small, with only 1.7M parameters, achieves a validation accuracy of 77.6%, surpassing the 38M parameter sparse convolution method MinkUNet, which achieves 73.3%.

## 5   Conclusion

**Conclusion.**   In this paper, we implement a point-based approach with linear complexity. Unlike current point-based methods, our framework uses space-filling curves to achieve neighbor query and downsampling with linear complexity. Additionally, we introduce a disassembled set aggregation module, which aggregates local features simply and elegantly, significantly reducing the redundant computations in neighborhoods and greatly enhancing scalability. Extensive experiments on multiple benchmarks demonstrate the efficiency and state-of-the-art performance of our method.

**Limitation and Future Work.** Although the proposed approach largely addresses the scalability challenges of point-based approaches in large-scale scenes, the distribution of point cloud data in memory in point-based approaches tends to be discontinuous, leading to inefficient memory access. This increases the cache miss rate, which in turn reduces the processing speed. Point-wise neighborhood aggregation also consumes a significant amount of memory. We hope that future work will address the significantly higher memory footprint than sparse convolution methods.

**Acknowledgment.** The authors would like to thank the reviewers of NeurIPS'24 for their constructive suggestions. This work was supported by the Key Research and Development Program of Shaanxi Province of China under Grant 2024GX-YBXM-149, in part by the Qinchuangyuan "Scientist + Engineer" Team Construction Project of Shaanxi Province of China under Grant 2022KXJ-009, in part by the National Natural Science Foundation of China under Grant 42271140, and in part by Northwest University Graduate Innovation Project under Grant CX2024194.

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

# Appendix

In the appendix, we provide more experiment details in Sec. A, and more experiment results in Sec. B.

## A   Experimental Details

This section provides a detailed description of the experimental setup for each dataset.

**Experimental environment.**

- CUDA version: 11.3
- PyTorch version: 1.12.1
- GPU: Nvidia RTX 4090D $\times$ 4
- CPU: AMD EPYC 9754 128-Core

**Training details.** The specific model training settings are shown in Tab. 8 and Tab. 9. We used cross-entropy loss in all experiments.

Table 8: Data augmentation.

|  | Rotate | Flip | Scale | Jitter | Chromatic Drop | Height | Grid Size |
|---|---|---|---|---|---|---|---|
| ScanObjectNN | ✓ |  | ✓ |  |  | ✓ |  |
| ModelNet40 |  |  | ✓ |  |  |  |  |
| S3DIS | ✓ |  | ✓ | ✓ | ✓ | ✓ | 0.04 |
| NuScenes | ✓ | ✓ | ✓ | ✓ |  |  | 0.05 |
| Sem. KITTI | ✓ | ✓ | ✓ | ✓ |  |  | 0.05 |

Table 9: Training setting.

|  | Epoch | Learning Rate | Weight Decay | Scheduler | Optimizer | Batch Size |
|---|---|---|---|---|---|---|
| ScanObjectNN | 250 | 0.002 | 0.05 | Cosine | AdamW | 32 |
| ModelNet40 | 600 | 0.001 | 0.05 | Cosine | AdamW | 32 |
| S3DIS | 3000 | 0.01 | 0.0001 | Cosine | AdamW | 8 |
| NuScenes | 50 | 0.002 | 0.05 | Cosine | AdamW | 12 |
| Sem. KITTI | 50 | 0.002 | 0.05 | Cosine | AdamW | 12 |

**Data license.** Our experiments use open-source datasets widely applied for 3D recognition research. The ScanObjectNN [53], SemanticKITTI [55], dataset is under the MIT license, while S3DIS [49], NuScenes [50], and ModelNet40 [54] have custom licenses that only allow academic use.

## B   Additional Quantitative Results

In this section, we present additional quantitative results of SemanticKITT [55] for 3D semantic segmentation. In addition, we provide semantic segmentation results for each category of NuScenes (see Tab. 10) and S3DIS Area 5 (see Tab. 11).

**SemanticKITTI.** The SemanticKITTI dataset consists of sequences from the original KITTI dataset, comprising a total of 22 sequences. Each sequence contains approximately 1,000 LiDAR scans, amounting to around 20,000 individual frames. The result is shown in Tab. 12. The mIoU of validation set and test set are 69.1% and 70.4% respectively.

**S3DIS 6-fold cross-validation.** To evaluate the generalization capabilities, we perform 6-fold cross-validation on the S3DIS dataset to ensure a robust assessment of our model's performance across different subsets of data. The results are shown in Fig. 13.

**Nomalization layer type.** We conducted ablation studies on the S3DIS dataset to further assess the necessity and effectiveness of BN. As shown in Tab. 14, models with BN outperform those with Layer Normalization (LN) and without any normalization, indicating that BN is particularly effective for our specific architecture.

Table 10: Semantic segmentation results on NuScenes val set. ‡ denotes using rotation and translation testing-time augmentations.

| Method | mIoU | barrier | bicycle | bus | car | construction | motorcycle | pedestrian | traffic cone | trailer | truck | driveable | other flat | sidewalk | terrain | manmade | vegetation |
|---|---|---|---|---|---|---|---|---|---|---|---|---|---|---|---|---|---|
| RangeNet53++ [36] | 65.5 | 66.0 | 21.3 | 77.2 | 80.9 | 30.2 | 66.8 | 69.6 | 52.1 | 54.2 | 72.3 | 94.1 | 66.6 | 63.5 | 70.1 | 83.1 | 79.8 |
| PolarNet [38] | 71.0 | 74.7 | 28.2 | 85.3 | 90.9 | 35.1 | 77.5 | 71.3 | 58.8 | 57.4 | 76.1 | 96.5 | 71.1 | 74.7 | 74.0 | 87.3 | 85.7 |
| Salsanext [37] | 72.2 | 74.8 | 34.1 | 85.9 | 88.4 | 42.2 | 72.4 | 72.2 | 63.1 | 61.3 | 76.5 | 96.0 | 70.8 | 71.2 | 71.5 | 86.7 | 84.4 |
| AMVNet [40] | 76.1 | 79.8 | 32.4 | 82.2 | 86.4 | **62.5** | 81.9 | 75.3 | **72.3** | **83.5** | 65.1 | **97.4** | 67.0 | **78.8** | 74.6 | 90.8 | 87.9 |
| Cylinder3D [41] | 76.1 | 76.4 | 40.3 | 91.2 | 93.8 | 51.3 | 78.0 | 78.9 | 64.9 | 62.1 | 84.4 | 96.8 | 71.6 | 76.4 | 75.4 | 90.5 | 87.4 |
| PVKD [39] | 76.0 | 76.2 | 40.0 | 90.2 | **94.0** | 50.9 | 77.4 | 78.8 | 64.7 | 62.0 | 84.1 | 96.6 | 71.4 | 76.4 | 76.3 | 90.3 | 86.9 |
| RPVNet [43] | 77.6 | 78.2 | 43.4 | 92.7 | 93.2 | 49.0 | 85.7 | 80.5 | 66.0 | 66.9 | 84.0 | 96.9 | 73.5 | 75.9 | 76.0 | 90.6 | 88.9 |
| 2DPASS [44] ‡ | 79.4 | 78.8 | 49.6 | 95.6 | 93.6 | 60.0 | 84.1 | 82.2 | 67.5 | 72.6 | 88.1 | 96.8 | 72.8 | 76.2 | **76.5** | 89.4 | 87.2 |
| SphereFormer [46]‡ | 79.5 | 78.7 | 46.7 | 95.2 | 93.7 | 54.0 | 88.9 | 81.1 | 68.0 | 74.2 | 86.2 | 97.2 | 74.3 | 76.3 | 75.8 | 91.4 | 89.7 |
| LinNet(ours) | 80.4 | 79.2 | 54.6 | 96.6 | 93.2 | 53.9 | 89.0 | 83.7 | 70.6 | 73.3 | 88.5 | 96.9 | 73.8 | 76.1 | 75.4 | 91.5 | 89.7 |
| LinNet‡(ours) | **81.4** | **80.0** | **56.9** | **96.9** | **94.0** | 58.4 | **90.0** | **84.4** | 72.1 | 74.2 | **89.7** | 97.0 | **74.4** | 76.8 | 76.0 | **91.7** | **89.9** |

Table 11: Semantic segmentation results on S3DIS Area 5.

| Method | OA | mAcc | mIoU | ceiling | floor | wall | beam | column | window | door | table | chair | sofa | bookcase | board | clutter |
|---|---|---|---|---|---|---|---|---|---|---|---|---|---|---|---|---|
| PointNet[7] | - | 49.0 | 41.1 | 88.8 | 97.3 | 69.8 | **0.1** | 3.9 | 46.3 | 10.8 | 59.0 | 52.6 | 5.9 | 40.3 | 26.4 | 33.2 |
| PointNet++[8] | 83.0 | - | 53.5 | - | - | - | - | - | - | - | - | - | - | - | - | - |
| PointCNN[14] | 85.9 | 63.9 | 57.3 | 92.3 | 98.2 | 79.4 | 0.0 | 17.6 | 22.8 | 62.1 | 74.4 | 80.6 | 31.7 | 66.7 | 62.1 | 56.7 |
| DGCNN[13] | 83.6 | - | 47.9 | - | - | - | - | - | - | - | - | - | - | - | - | - |
| DeepGCN[51] | - | - | 52.5 | - | - | - | - | - | - | - | - | - | - | - | - | - |
| KPConv[33] | - | 72.8 | 67.1 | 92.8 | 97.3 | 82.4 | 0.0 | 23.9 | 58.0 | 69.0 | 81.5 | 91.0 | 75.4 | 75.3 | 66.7 | 58.9 |
| ASSANet-L[18] | - | - | 66.8 | - | - | - | - | - | - | - | - | - | - | - | - | - |
| Point Trans.[9] | 90.8 | 76.5 | 70.4 | 94.0 | 98.5 | **86.3** | 0.0 | 38.0 | 63.4 | 74.3 | 82.4 | 89.1 | 74.3 | 80.2 | 76.0 | 59.3 |
| RepSurf-U[57] | 90.2 | 76.0 | 68.9 | - | - | - | - | - | - | - | - | - | - | - | - | - |
| PointVector[57] | 91.0 | 78.1 | 72.3 | **95.1** | **98.6** | 85.1 | 0.0 | 41.4 | 60.8 | 76.7 | 84.4 | 92.1 | 82.0 | 77.2 | **85.1** | 61.4 |
| PointNeXt [22] | 90.7 | 77.5 | 70.8 | 94.2 | 98.5 | 84.4 | 0.0 | 37.7 | 59.3 | 74.0 | 83.1 | 91.6 | 77.4 | 77.2 | 78.8 | 60.6 |
| **LinNet(ours)** | **91.9** | **79.0** | **73.7** | 94.8 | 98.5 | 86.2 | 0.0 | **45.5** | 61.6 | **82.8** | **85.1** | **92.3** | **85.5** | **80.0** | 80.4 | **65.4** |

Table 12: Sem. seg. on Sem. KITTI.

| Methods | Val | Test |
|---|---|---|
| SPVNAS [42] | 64.7 | 66.4 |
| Cylinder3D [41] | 64.3 | 67.8 |
| PVKD [39] | - | 71.2 |
| 2DPASS [44] | 69.3 | 72.9 |
| WaffleIron [47] | 68.0 | 70.8 |
| SphereFormer [46] | 67.8 | 74.8 |
| RangeFormer [45] | 67.6 | 73.3 |
| MinkUNet [4] | 63.8 | - |
| LinNet (ours) | 69.1 | 70.4 |

Table 13: S3DIS 6-fold cross-validation.

| Methods | mIoU (%) | mAcc (%) | OA (%) |
|---|---|---|---|
| PointNeXt | 74.9 | 83.0 | 90.3 |
| LinNet | 78.6 | 86.3 | 91.9 |

Table 14: Normalization layer.

| None | BN | LN |
|---|---|---|
| 71.8% | 72.9% | 71.9% |

