# OpenReview forum: "LinNet: Linear Network for Efficient Point Cloud Representation Learning"
_NeurIPS.cc/2024/Conference — NeurIPS 2024 poster_

### Official Review · Reviewer_zyd3 · 2024-07-05

**Soundness:** 4
**Presentation:** 4
**Contribution:** 4
**Rating:** 8
**Confidence:** 4

**Summary:**

The submission #308 entitled "LinNet: Linear Network for Efficient Point Cloud Representation Learning" introduces a linear network designed for efficient point cloud representation learning. To achieve this task, the authors propose a novel disassembled set abstraction (DSA) module and a linear sampling strategy, which together enhance computational efficiency and scalability. The method maps 3D point clouds onto 1D space-filling curves, allowing for parallelization of downsampling and neighborhood queries on GPUs with linear complexity. This approach achieves state-of-the-art performance across various benchmarks.

**Strengths:**

- The linear sampling strategy is very elegant.
- The large-scale comparison across multiple datasets and many approaches is highly appreciable.
- The authors offer to open their code upon acceptance.
- The ablation studies are well-conducted, clearly showing the contribution of each proposed module.
- The reasoning behind DSA is well introduced and motivated.
- The editorial quality of the paper is excellent. It is easy to read, and the illustrations are pleasant and informative.
- The assessments are numerous and conclusive, showing that the proposed approach is innovative and achieves state-of-the-art results on multiple metrics.

**Weaknesses:**

- One negative point is the significant memory footprint of the approach, as mentioned in the limitations section.

**Questions:**

- Would it be possible to conduct a small experiment to measure the memory footprint of the approach?
- It would be interesting to have a few cross-validation tests to know more about the generalization of the technique.

**Limitations:**

The limitations of the approach are well covered at the end of the manuscript.

---

> ### Author Rebuttal · Authors · 2024-08-07
>
> We sincerely thank you for your time and constructive comments. In the following, we address your concerns carefully.
>
> **W1: One negative point is the significant memory footprint of the approach, as mentioned in the limitations section.**
>
> **A:** Thank you for addressing the concern regarding the memory footprint of our approach, which we have acknowledged in the "Limitations" section of our manuscript.
>
> While our local aggregation methods have indeed improved the scalability of point-based approaches in large-scale scenarios by reducing memory usage, the overall memory footprint remains a substantial challenge. We are actively working on strategies to further minimize memory consumption, which we plan to implement in future iterations of our model.
>
> Firstly, we plan to integrate hash query strategies with our local aggregation methods. Employing hash tables to build rulebooks, similar to those used in sparse convolutions, will allow us to manage memory more efficiently.
>
> Secondly, we aim to make better use of shared memory on GPUs to enhance processing efficiency. Shared memory is much faster than global memory and using it effectively can greatly reduce the need for frequent global memory accesses, which are more costly in terms of time and energy consumption.
>
> -----
>
> **Q1: Experiment to measure the memory footprint of the approach**.
>
> **A:** Thank you for your constructive suggestion regarding the measurement of our model's memory footprint. To address this, we conducted experiments to evaluate memory usage during both training and inference phases on the NuScenes dataset, utilizing an RTX 4090 graphics card with all tests conducted at a batch size of 1.
>
> We included comparisons with the baseline model PointNeXt [1] and the sparse convolution method MinkUNet [2]. Our findings reveal that PointNeXt suffers from out-of-memory issues when handling large-scale scenes, highlighting scalability challenges. In contrast, our DSA module significantly reduces memory consumption by avoiding high-dimensional feature transformations on neighboring point clouds.
>
> Given that MinkUNet starts with 32 input channels, we conducted similar tests with our LinNet-Small model, which also has 32 initial feature channels, for a direct comparison:
>
> |              | Training Mem. (NuScenes) | Inference Mem. (NuScenes) |
> | ------------ | ------------------------ | ------------------------- |
> | MinkUNet     | 2.6 GB                   | 1.4 GB                    |
> | PointNeXt    | Out of Memory            | Out of Memory             |
> | LinNet-Small | 5.2 GB                   | 4.9 GB                    |
> | LinNet       | 16 GB                    | 13 GB                     |
>
> Although LinNet-Small consumes more memory than MinkUNet, it is crucial to note that LinNet-Small, with only 1.7M parameters, achieves a validation accuracy of 77.6%, surpassing the 38M parameter sparse convolution method MinkUNet, which achieves 73.3%. This demonstrates that our model, despite its higher memory footprint, provides superior accuracy, offering a significant advantage in scenarios where performance is critical.
>
> -----
>
> **Q2: Conduct cross-validation.**
>
> **A:** Thank you for suggesting the inclusion of cross-validation tests to evaluate the generalization capabilities of our technique. We have performed 6-fold cross-validation on the S3DIS dataset to ensure a robust assessment of our model's performance across different subsets of data. Here are the results:
>
> | Methods   | mIoU (%) | mAcc (%) | OA (%) |
> | --------- | -------- | -------- | ------ |
> | PointNeXt | 74.9     | 83.0     | 90.3   |
> | LinNet    | 78.6     | 86.3     | 91.9   |
>
> These results demonstrate that LinNet consistently outperforms the baseline model, PointNeXt, across multiple metrics, which suggests superior generalization abilities.
>
> -----
>
> [1] Qian et al. PointNeXt: Revisiting PointNet++ with Improved Training and Scaling Strategies. NeurIPS 2022.
>
> [2] Choy et al. Minkowski convolutional neural networks. CVPR 2019.
>
> ------
>
> We hope our response adequately addresses your concerns. If you still have any questions, we are looking forward to hearing them.

---

> > ### Comment · Reviewer_zyd3 · 2024-08-13
> >
> > I would like to thank the authors for their clarifications. After reading the comments from other reviewers, I realize I may have been slightly too generous in my initial assessment. However, I still believe this manuscript is worthy of appearing in NeurIPS, and I would like to maintain my initial rating.

---

> > > ### Author Response · Authors · 2024-08-14
> > >
> > > Dear Reviewer zyd3,
> > >
> > > Thanks for your constructive suggestions.
> > >
> > > We will improve our paper's quality based on your guidance and comments.
> > >
> > > Thank you for recognizing our work! We hope our simple yet effective LinNet would help the community towards a better understanding of point cloud analysis.
> > >
> > > Best, Authors

---

### Official Review · Reviewer_aBR6 · 2024-07-09

**Soundness:** 3
**Presentation:** 3
**Contribution:** 3
**Rating:** 6
**Confidence:** 3

**Summary:**

In this work, the authors propose an efficient learning framework for point cloud representation learning. For the computational intensive local aggregation operation, this work proposes Disassembled set abstraction (DSA) to aggregate local features in terms of the spatial distributions of points in a simple and efficient manner. This work also proposes a Linearization sampling strategy and hash query operation to accelerate the sampling and neighbor searching processes. Experiments on classification and semantic segmentation demonstrate the effectiveness of the proposed method.

**Strengths:**

1. The proposed local aggregation operation seems to be simple, efficient, and effective;

2. The point searching strategy including the Linearization sampling strategy and hash query can indeed improve the effciency while keeping the overall performances;

**Weaknesses:**

My major concern about this work is the higher performances than transformer-based methods. It may be a little hard to understand why the proposed DSA and hash-based searching operations can improve the performances so greatly. Because these operations are more like the approximation of corresponding operations in PointNet++.

**Questions:**

1. Why do we introduce batch normalization to the aggregated features in Eq.4? As the addition of features and neighborhood features have already introduce the spatial characteristics, I am not sure if this BN is necessary here, or other simple components may also work;

2. In the Hash query part, will the points in a same local grid share the same neighbors? I do not quite get the calculation of complexity for each point.

3. Is the hash grid is pre-constructed before training? Or created repeatedly during training?

3. From the results in Table 1 and Table 3, the proposed method even outperforms transformer-based methods in a more efficiency way. Could the author analyze the reasons behind this? As the improvements of this framework seem to be efficient simplification of existing PointNet++ framework, I am curious why it can improve the performances so obviously.

Please check the grammar also, e.g., In Line 101, the $x_i$ might actually be $p_i$;

**Limitations:**

Yes.

---

> ### Author Rebuttal · Authors · 2024-08-07
>
> We sincerely thank you for your time and constructive comments. In the following, we address your concerns carefully.
>
> **W1 & Q4: Why the proposed DSA and hash-based searching operations can improve the performances so greatly? (Higher than transformer-based methods)**
>
> **A:** Thank you for your insightful concerns regarding the superior performance. It is important to clarify that though our method is built upon the PointNet++ style framework, LinNet is not a superficial approximation but a substantial exploration and design to specifically address scalability and efficiency. Our model, LinNet, addresses a longstanding issue with the PointNet++ style framework: the lack of scalability.
>
> We try to explain the significant performance improvements from two aspects.
>
> **(a) Comparison with PointNet++ style framework**
>
> - **Accuracy**: Fig. 4 of the manuscript illustrates that after several training epochs, both the DSA and vanilla SA stabilize in terms of loss; however, DSA consistently maintains a lower loss compared to vanilla SA. This considerable difference demonstrates that vanilla SA may not sufficiently adapt to the datasets, whereas DSA exhibits a more robust capability to fit the data, thereby achieving higher accuracy.
>
> - **Speed**: In DSA, point-wise convolutions are applied directly to anchor features rather than to an expansive neighborhood, thus **DSA requires significantly fewer FLOPs compared to vanilla SA**. Additionally, the adoption of a linear complexity search strategy shifts away from the costly traditional algorithms like FPS and KNN—which have complexities of $\mathcal{O}(N^2)$ and $\mathcal{O}(kN^2)$ respectively—to more efficient, GPU-friendly linear complexity algorithms. This enhances computational efficiency remarkably.
>
> **(b) Comparison with Transformer-based method.**
>
> The Point Transformer v2 (PTv2) we are comparing is also essentially a point-based approach, which implements downsampling through grid pooling similar to the resize operation in image and uses local attention for feature aggregation.
>
> - **Accuracy**: We observe that the limited receptive field during the downsampling stages of PTv2 and similar methods could restrict their accuracy. PTv2 samples only within a single grid (typically about 6 points), whereas our method captures the nearest $k$ points from the target grid and its 26 surrounding grids, providing a broader receptive field.
> - **Speed**: Although PTv2's grid sampling avoids the need for farthest point sampling, its reliance on KNN for neighborhood queries and the complexity of its local attention calculations significantly hamper its processing speed. In contrast, our DSA module simplifies these processes, leading to faster data processing.
>
> -----
>
> **Q1: Why adapted BN in Eq. (4). Is this BN necessary, or other simple components also work?**
>
> **A:** Thank you for your interest in batch normalization. As you mentioned, the addition has already introduced spatial characteristics. Note that BN is placed **after the max pooling layer to normalize the aggregated feature**, facilitating subsequent convergence. In the ScanObjectNN dataset, BN is necessary and has led to a 0.4% OA improvement. As you suggested, we conducted ablation studies on the S3DIS dataset to further assess the necessity and effectiveness of BN. Here are the results averaged over three experiments:
>
> | None   | BN     | LN     |
> | ------ | ------ | ------ |
> | 71.8 % | 72.9 % | 71.9 % |
>
> As shown, models with BN outperform those with Layer Normalization (LN) and without any normalization, indicating that BN is particularly effective for our specific architecture.
>
> -----
>
> **Q2: In the Hash query part, will the points in the same local grid share the same neighbors?**
>
> **A:** Thank you for your interest in the hash query part.  Our hash query confines the search range of each point to its own grid and the adjacent 26 neighborhood grids (i.e., $3\*3\*3-1$), selecting the closest $k$ points as neighbors from these grids. **Although points in the same local grid have an identical search range, the specific neighbors selected for each point (i.e., the $k$ nearest points) can differ.** This variation arises because **neighbor points are chosen based on the actual spatial distances between points,** not merely by their presence in the same grid.
>
> Regarding the computation complexity discussed in the manuscript, we apologize for any confusion caused by omitting the complexity of heap sorting. For a point cloud comprising $N$ points distributed across $m$ non-empty grids, constructing the hash table entails a complexity of $\mathcal{O}(m)$. Assuming each point’s 27-grid neighborhood contains $p$ points on average, identifying the closest $k$ points involves maintaining a heap with a complexity of $\mathcal{O}(p \log k)$ and a final sorting step costing $\mathcal{O}(k \log k)$. Thus, the total computational complexity is $\mathcal{O}(m + N(p \log k + k \log k))$. We will clarify and elaborate on these calculations in the revised version of our manuscript to prevent confusion.
>
> -----
>
> **Q3: Is the hash grid is pre-constructed before training? Or created repeatedly during training?**
>
> **A:** Thank you for your question regarding the hash table. In our approach, the hash table is not pre-constructed but is instead dynamically built-in real-time during the training process. This allows for hash query and linearization sampling to be integrated seamlessly into each training iteration. As demonstrated in Fig. 1, both operations are efficiently parallelized on the GPU and collectively account for less than 10% of the model's forward time.
>
> -----
>
> **Q4: The typo in line 101.**
>
> **A:** Thank you for pointing out this typo. You are correct that the symbol "$x_i$" should indeed be "$p_i$". We have made this correction in the revised manuscript.
>
> ------
>
> We hope our response adequately addresses your concerns. If you still have any questions, we are looking forward to hearing them.

---

> > ### Comment · Reviewer_aBR6 · 2024-08-10
> > **Response**
> >
> > Thank you for your rebuttal. It has addressed most of my concerns. However, I am still curious about the reason why DSA outperforms SA. Although Fig.4 confirms that DSA can converge to lower loss than SA, DSA seems to be a more efficient simplification of SA. Could you provide some more intuitive explanation about the reasons behind such improvements? That is, why DSA exhibits a more robust capability to fit the data than SA?

---

> ### Author Response · Authors · 2024-08-11
>
> **A:** Thank you for your continued interest in the Disassembled Set Abstraction (DSA) and Set Abstraction (SA). Your question regarding why DSA outperforms SA in terms of data fitting is insightful. I'll provide a more intuitive explanation focusing on the architectural differences and their impacts.
>
> In a nutshell, the **DSA module places more emphasis on the extraction of geometric information which is crucial for point cloud learning**.
>
> For SA, the feature of an anchor is updated as:
> $$
> \mathbf{f}_i' = \mathcal{R} _{j:(i, j)\in \mathcal{N}} \\{\text{PWConv}^{3+c \mapsto c}(\mathbf{f}_j||(\mathbf{p}_j-\mathbf{p}_i))\\}.
> $$
>
> For DSA, the process is:
> $$
> \mathbf{\overline{f}}_i  = \text{PWConv}^{c \mapsto c}(\mathbf{f}_i);
> \mathbf{f}_i' = \text{BN} \\{{\mathcal{R} _{j:(i, j)\in \mathcal{N}} \{\overline{\mathbf{f}}_j + \text{PWConv}^{3 \mapsto c}((\mathbf{p}_j-\mathbf{p}_i))} }\\}.
> $$
>
> Excluding the Batch Normalization and focusing only on neighbor feature computations for simplicity, let $\mathbf{y}_j=[y_j^1, y_j^2,..., y_j^c]$ represent **the features of the $j$-th neighbor**. Treating pointwise convolution as a linear layer and without considering a bias, the SA model uses a weight matrix $\mathbf{W}$ of dimensions $c \times (c+3)$ to process both semantic and geometric information concurrently. The input $\mathbf{x}_j = [\mathbf{f}_j, \Delta\mathbf{p}_j]$ includes semantic features $\mathbf{f}_j$ (dimension $c$) and geometric features $\Delta\mathbf{p}_j$ (dimension $3$), with the output defined as $\mathbf{y}_j = \mathbf{W}\mathbf{x}_j^\text{T}$. The output for the $k$-th channel is given by:
> $$
> y_j^k = [w^{k1}, w^{k2}, \ldots, w^{kc}, w^{k(c+1)}, w^{k(c+2)}, w^{k(c+3)}][\mathbf{f}_j, \Delta\mathbf{p}_j]^\text{T}.
> $$
> With Kaiming initialization, the weight matrix $\mathbf{W}$ is initialized to a normal distribution $\mathcal{N}(0, \sqrt{\frac{2}{c+3}})$. This initialization ensures uniformity across all weights, meaning the weights for geometric inputs contribute $\frac{3}{c+3}$ to the total output. Consequently, the influence of geometric information on the overall results is significantly limited.
>
> In contrast, the DSA model separates the processing of semantic and geometric information through two distinct linear layers. It incorporates two weight matrices, $\mathbf{W}_f$ and $\mathbf{W}_p$, corresponding to the dimensions $c \times c$ and $c \times 3$, respectively. The output is determined by:
> $$
> \mathbf{y}_j = \mathbf{W}_f \mathbf{f}_j + \mathbf{W}_p \Delta\mathbf{p}_j.
> $$
> For the $k$-th channel, the output is:
> $$
> y_j^k = [w_f^{k1}, w_f^{k2}, \ldots, w_f^{kc}]\mathbf{f}_j^\text{T} + [w_p^{k1}, w_p^{k2}, w_p^{k3}]{\Delta\mathbf{p}_j}^\text{T},
> $$
> where $\mathbf{W}_f \sim \mathcal{N}(0, \sqrt{\frac{2}{c}}),\mathbf{W}_p \sim \mathcal{N}(0, \sqrt{\frac{2}{3}})$ . This initialization strategy enables the DSA module to appropriately tailor the weights based on the number of input channels. Although there are only three channels dedicated to geometric information, their relatively larger weights enhance the model's capability to more effectively extract geometric information.
>
> In summary, DSA and SA are equivalent in terms of mathematical expression during forward propagation. In SA, a single linear layer merges $\mathbf{f}$ and $\Delta\mathbf{p}$, transforming them linearly to produce the output. In contrast, DSA employs two distinct linear layers to separately process $\mathbf{f}$ and $\Delta\mathbf{p}$, and then sums the outputs. However, the different weight initialization of the two linear layers causes the network to preferentially learn from geometric information. This bias enables the network to more effectively detect patterns associated with geometry, which is particularly advantageous in point cloud analysis.
>
> Thank you once again for your insightful suggestions, which have prompted us to further explore the underlying logic. We hope our response adequately addresses your concerns. If you still have any questions, we are looking forward to hearing them.

---

> > ### Comment · Reviewer_aBR6 · 2024-08-11
> >
> > Thanks for your responses. The initialization could be a potential reason. Considering its good overall performances, I will raise my score to weak accept.

---

> > > ### Author Response · Authors · 2024-08-11
> > >
> > > Dear Reviewer aBR6,
> > >
> > > Thanks for your constructive suggestions and the score increase.
> > >
> > > We will improve our paper's quality based on your guidance and comments.
> > >
> > > We hope our simple yet effective LinNet would help the community towards a better understanding of point cloud analysis.
> > >
> > > Best, Authors

---

### Official Review · Reviewer_7sY3 · 2024-07-12

**Soundness:** 4
**Presentation:** 3
**Contribution:** 3
**Rating:** 6
**Confidence:** 4

**Summary:**

The paper proposes a novel lightweight backbone network model for input point cloud data suitable for global and local per-point feature extraction. It relies on two main ideas: (1) the separate processing of point coordinates and features (and a further combination of these two streams of features before the neighborhood pooling operation), (2) the use of the space-filling curves to define local neighborhoods that allow hash queries and linear complexity sampling.

The approach is evaluated in the point cloud classification and segmentation tasks on ModelNet40, ScanObjectNN and S3DIS, NuScenes datasets respectively, demonstrating performance competitive to state of the art. Additional experiments include ablations exploring the efficacy of every proposed component and comparison to other methods in terms of model efficiency.

**Strengths:**

* The proposed feature aggregation method is likely novel and improves the results according to the ablation studies.
* The proposed method is on par with the state-of-the-art non-transformer-based approaches but is better in terms of scalability to larger point clouds.
* The extensive evaluation shows the importance of every component in the ablations.

**Weaknesses:**

* All the improvements (except for the NuScenes dataset) are not particularly distinctive.
* While present the efficiency is not exploited in any presented applications (for classification point clouds are small, so other methods work fast as well, for segmentation there are no efficiency comparisons).
* The text is written well but some figures can be improved.

**Questions:**

The proposed method relies on the space-filling curves introduced in PointTransformer v.3, which is mentioned multiple times in the paper. At the same time, it is not considered for comparison. While PTv3 is concurrent, for completeness, it would be nice to include the results from it, especially since this work already acknowledges the existence of PTv3. So my question is how this method compares to PTv3 in terms of performance and efficiency?

Figure 5 shows a nice uniformly distributed (over the grid) point cloud but in practice, any grid-based discretization suffers to some extent from the discretization artifacts. The proposed method can in principle have cells with single points. Do these cells exist in practice and do they cause any problems?

Figure 1: a) and b) have the same area but the total time of b) is significantly lower which is misleading.

Figure 3: The choice of the number of inputs, intermediate features, and outputs is either arbitrary or not clear which is confusing. Showing the operations for a single input point would be clearer.

Figure 5: Showing an empty table in c) is not informative. Showing a part of the actual hash table for this example in the figure might work better.

**Limitations:**

The authors properly address the limitations in the submitted draft.

---

> ### Author Rebuttal · Authors · 2024-08-07
>
> We sincerely thank you for your time and constructive comments. In the following, we address your concerns carefully.
>
> **W1: The improvements (except for the NuScenes dataset) are not particularly distinctive.**
>
> **A:**   Thank you for your comments.
>
> Firstly, for small-scale classification tasks, we employed the same training protocols and experimental conditions as the SOTA benchmark, PointNeXt [1]. Nonetheless, we still achieve an overall accuracy (OA) improvement of 0.4% on the ScanObjectNN dataset, while other PointNeXt style architectures (e.g., PointVector [2], PointMetaBase [3]) using the same experimental setup only achieve an OA improvement of 0.1%.
>
> Secondly, for large-scale segmentation tasks, except for the NuScenes dataset, we also achieved a 2.5% improvement in mIoU without Test Time Augmentation (TTA) and a **2.9% improvemen**t with TTA. The improvements are particularly impressive and distinctive. **The speedup is even more impressive than the accuracy improvement**, and **the magnitude of the speedup increases with the size of the point cloud**. Moreover, **the enhancement in processing speed**—a critical factor for large-scale applications—**is even more striking**. Specifically, at a 20k point cloud size, our inference speed is **1.7 times faster** than PointNeXt (47ms vs. 87ms), scaling up to a speedup ratio of **13 times** (232 ms vs. 3147ms) when handling 200k points.
>
> **This dual achievement of improved speed and accuracy, especially at larger scales, is noteworthy**.
>
> -----
>
> **W2: The efficiency is not exploited in any presented applications.**
>
> **A:** Thank you for pointing out this issue. Visual comparisons of model efficiency at four different point cloud sizes (20k, 50k, 100k, 200k) are shown in Fig. 7(a). Following your suggestion, we further add a table to more clearly and intuitively demonstrate efficiency. Corresponding to the number of points pre-sampled by PointNeXt on the S3DIS dataset, we replaced 20k with 24k. Thanks to the simplicity and efficiency of our DSA module, the proposed LinNet exhibits only half the latency of Point Transformer v2 [4]. Remarkably, at the 200k level, our LinNet model performs 13 times faster than PointNeXt.
>
> Model latency on different scales (ms):
>
> | Methods   | 24 k | 50 k | 100 k | 200 k |
> | --------- | ---- | ---- | ----- | ----- |
> | PointNeXt | 87   | 266  | 878   | 3147  |
> | PTv2      | 55   | 163  | 228   | 493   |
> | LinNet    | 47   | 62   | 123   | 232   |
>
> -----
>
> **W3: Some figures can be improved.**
>
> **A:** Thank you for the constructive suggestions and valuable feedback.
>
> The revised figure can be found in the **PDF** of the global rebuttal. Fig. 1, Fig. 2, and Fig. 3 of the global rebuttal correspond to Fig. 1, Fig. 3, and Fig. 5 of the manuscript.
>
> - **Figure 1:** We have redrawn Fig. 1 (a) and added the total inference latency to it.
> - **Figure 3:** We are sorry for this confusion. Following your suggestion, we take a single point as input and set the number of neighbors to 3. We represent the features using rectangles of uniform size to maintain consistency across the data representation.
> - **Figure 5:** Thank you for your insightful feedback on Fig. 5, and your concerns about artifacts from grid-based discretization. Given the sparse nature of the data distribution and the characteristics of the discretization process, our method accommodates scenarios where a grid may contain only a single point. This point is then directly used as a new sampling point under our strategy, ensuring that the distribution of the newly sampled point cloud closely mirrors that of the original. Following your suggestions, we have made specific optimizations in the revised version to ensure that the number of points in each grid varies, better reflecting the variability found in real-world data distributions. Additionally, in Fig.5 (c), we have incorporated actual data into the hash table.
>
> -----
>
> **Q1: Without result of Point Transformer v3**.
>
> **A:** Thank you for pointing out this issue. A detailed comparison with PointTransformerv3 (PTv3) is available in the comments below. Our model consistently achieves better results than PTv3 on the S3DIS and NuScenes validation sets, also demonstrating competitive performance on NuScenes testing sets.
>
> Regarding latency, PTv3 utilizes spatial curves to divide the point cloud into patches. This allows the model to compute attention on the patches rather than on individual points, avoiding the need for point-wise local attention. In contrast, our method leverages point-wise local features, resulting in higher latency compared to PTv3. However, we would like to highlight that our LinNet is a pure MLP network and does not require any additional operational assistance. Instead, PTv3 relies on sparse convolution for positional encoding and the sparse convolution kernel incurs more model parameters.
>
> Model performance:
>
> |        | S3DIS Area 5 | S3DIS 6-fold | NuScenes (val) | NuScenes (test) |
> | ------ | ------------ | ------------ | -------------- | --------------- |
> | PTv3   | 73.4         | 77.4         | 80.4           | 82.7            |
> | LinNet | 73.7         | 78.6         | 81.4           | 82.3            |
>
> Model size and execution time, latency are measured with 24k points:
>
> |        | Model Size (M) | Forward Latency (ms) |
> | ------ | -------------- | -------- |
> | PTv3   | 46.2           | 26                   |
> | LinNet | 14.7           | 47                   |
>
> -----
>
> [1] Qian et al. PointNeXt: Revisiting PointNet++ with Improved Training and Scaling Strategies. NeurIPS 2022.
>
> [2] Deng et al. PointVector: A Vector Representation In Point Cloud Analysis. CVPR 2023.
>
> [3] Lin et al. Meta Architecture for Point Cloud Analysis. CVPR 2023.
>
> [4] Wu et al. Point Transformer V2: Grouped Vector Attention and Partition-based Pooling. NeurIPS 2022.
>
> ------
>
> We hope our response adequately addresses your concerns. If you still have any questions, we are looking forward to hearing them.

---

### Official Review · Reviewer_J8xV · 2024-07-20

**Soundness:** 3
**Presentation:** 2
**Contribution:** 2
**Rating:** 4
**Confidence:** 3

**Summary:**

This paper proposes a method for point cloud segmentation and classification. The main contribution is making the local aggregation dependent on the anchor point. The approach demonstrates improvements of one to two percent on S3DIS and NuScenes datasets compared to existing methods.

**Strengths:**

- The architecture is based purely on an MLP (with some hashing operations) which is a significant benefit in terms of implementation and potentially in terms of computational efficiency.
- The approach of making local aggregation anisotropic and the proposed DSA module has a good motivation.
- The method shows improved performance on standard benchmarks (S3DIS and NuScenes), outperforming existing approaches by a small but consistent margin.

**Weaknesses:**

- The paper's writing seems overly complex with lots of unnecessary jargon making it difficult to follow the core ideas and contributions.
- The improvements in performance, while consistent, are relatively small (1-2%), which raises questions about the practical significance of -the method.
- The modifications proposed are individually well-motivated but seem somewhat ad hoc, lacking a strong theoretical foundation.

**Questions:**

1. Have you explored whether a more general model could learn the invariances you've built into LinNet without explicit architectural choices?
2. Given the relatively small improvements in accuracy, what do you see as the main practical advantages of LinNet over existing methods?

**Limitations:**

The discussion of limitations is sufficient.

---

> ### Author Rebuttal · Authors · 2024-08-07
>
> We sincerely thank you for your time and constructive comments. In the following, we address your concerns carefully.
>
> **W1: The paper's writing seems overly complex with lots of unnecessary jargon making it difficult to follow the core ideas and contributions.**
>
> **A:** We sincerely apologize for any confusion caused by the writing complexity of our manuscript and appreciate your feedback on the use of jargon. In response, we have revised the manuscript to simplify the language and ensured that key concepts are explained more thoroughly. For example, in the context of DSA module, we use terms like **anisotropy** and **isotropy**. To clarify, we employ the term **spatial-wise anisotropy** to refer to variations among the features of neighboring points within the same feature channel, and **channel-wise anisotropy** to describe differences across various feature channels.
>
> The core idea of our paper is to **enhance the scalability of existing point-based methods through a more lightweight feature aggregation strategy and a point cloud search strategy with reduced linear complexity**.
>
> -----
>
> **W2: The performance improvements (1-2%) are relatively small, which raises questions about the practical significance.**
>
> **A:** Firstly, **in the domain of point cloud processing, even seemingly modest improvements of 1-2% are indeed substantial.** For instance, Point Transformer [1] (first released in Dec 2020)  achieved a result of 70.4%, while the result of Point Transformer V2 [2] (Oct 2022) is 71.6%.
>
> Secondly, we want to highlight that our approach **achieves dual improvements in both speed and accuracy**. Previous work, such as the Fast Point Transformer [3], sacrifices accuracy in the pursuit of efficiency. In contrast, we achieve a 2.9% mIoU improvement while much faster than PointNeXt [5] on S3DIS. More critically, by employing a linear sampling strategy and a DSA module, **we have largely addressed the long-standing scalability challenges associated with point-based networks.** This allows our network to be easily applied to large-scale point cloud scenes. We believe these aspects affirm the practical significance of our reported improvements, offering meaningful contributions to the field.
>
> -----
>
> **W3: The modifications proposed are individually well-motivated but seem somewhat ad hoc, lacking a strong theoretical foundation.**
>
> **A:** Thank you for your endorsement of the motivation for our work.
>
> **Our modifications are not ad hoc but are derived from a thorough analysis and adaptation of established concepts within both 2D and 3D vision technologies.** To improve the efficiency of point-based networks, we started by systematically reviewing local aggregation techniques used in computer vision. Inspired by the success of separable convolutions in 2D imaging, we explored their applicability to 3D point clouds.
>
> **We conducted a detailed theoretical analysis of the set abstraction (SA) module used in 3D vision, examining it from the perspective of anisotropy in Sec. 3.2.** It shows that principles of separable convolution, effective in 2D vision, could be adapted for 3D point clouds, thus motivating our modifications. Initially, we attempted to directly separate spatial anisotropy and channel anisotropy from the input features. However, the initial experimental outcomes were not as anticipated, which prompted further in-depth analysis.
>
> This led to the development of the DSA module, where we refined our approach based on our theoretical insights and experimental findings. **Finally, we validated our modifications with explanatory analysis and empirical evidence, clearly demonstrating the advantages of the DSA module over the vanilla SA.**
>
> Following your feedback, we will further **strengthen the theory by incorporating references such as ASSANet [4], which provides an in-depth theoretical analysis of anisotropy in point cloud processing.**
>
> -----
>
> **Q1: Have you explored whether a more general model could learn the invariances you've built into LinNet without explicit architectural choices?**
>
> **A:** Thank you for your insightful question about exploring a more generalized model without explicit architectural choices.
>
> In the manuscript, for the segmentation used in S3DIS, the number of layers of our model was set to [4, 7, 4, 4] for a fair comparison with the baseline PointNeXt . For the other dataset, NuScenes, which is much larger, we adjusted the number of layers in the encoder part to [4, 4, 7, 4] for efficiency. In response to your question, we used a uniform [4, 7, 4, 4] configuration across tasks to determine if a more generalized structure would impact performance. The results show that the mIoU of NuScenes decreases by only 0.2% (80.2% vs. 80.4%), which indicates that the model performs robustly in both configurations.
>
> -----
>
> **Q2: Apart from accuracy, the practical advantages of LinNet over existing methods.**
>
> **A:** Thank you for your question.  In addition to improved accuracy,  **the substantial increase in processing speed represents a significant practical benefit of our approach, particularly for large-scale applications**. As demonstrated in Fig. 7(a), LinNet achieves a remarkable improvement in response speed; when handling point clouds of up to 200k points, it performs **13 times faster** than the current state-of-the-art method, PointNeXt++, and **twice as fast as PTv2**.
>
> -----
>
> [1] Zhao et al. Point Transformer. ICCV 2019.
>
> [2] Wu et al. Point Transformer V2: Grouped Vector Attention and Partition-based Pooling. NeurIPS 2022.
>
> [3] Park et al. Fast Point Transformer. CVPR 2022.
>
> [4] Qi et al. ASSANet: An Anisotropic Separable Set Abstraction for Efficient Point Cloud Representation Learning. NeurIPS 2021.
>
> [5] Qian et al. PointNeXt: Revisiting PointNet++ with Improved Training and Scaling Strategies. NeurIPS 2022.
>
> ------
>
> We hope our response adequately addresses your concerns. If you still have any questions, we are looking forward to hearing them.

---

> ### Author Response · Authors · 2024-08-11
> **Supplement on weakness 3**
>
> Inspired by the comment from reviewer aBR6, we delve deeper into the reason why DSA performs better than SA from the standpoint of parameter initialization.
>
> To clarify, we replace pointwise convolutions with linear layers and restructure both the DSA and SA modules.
> In SA, a single linear layer merges the neighborhood features $\mathbf{f}_j$ and the positional differences $\Delta\mathbf{p}_j$, transforming them linearly to produce the output as follows:
> $$
> \mathbf{f}_i' = \mathcal{R} _{j:(i, j)\in \mathcal{N}} \\{\text{Linear}^{3+c \mapsto c}([\mathbf{f}_j, \Delta\mathbf{p}_j])\\}.
> $$
>
> Contrastingly, DSA utilizes two separate linear layers to process $\mathbf{f}_j$ and $\Delta\mathbf{p}_j$ individually and then combines the outputs. **Ignoring the removal of redundant calculations in the neighborhood and Batch Normalization (BN)**, the DSA module is depicted as:
>
> $$
> \mathbf{f}_i' = {\mathcal{R} _{j:(i, j)\in \mathcal{N}} \\{ \text{Linear}^{c \mapsto c}(\mathbf{f}_j) + \text{Linear}^{3 \mapsto c}(\Delta\mathbf{p}_j)} \\}.
> $$
>
> Both modules appear mathematically equivalent during forward propagation. Yet, **distinct initializations of the two linear layers in DSA encourage the network to prioritize geometric information more effectively.** Excluding bias, the SA module uses a combined weight matrix $\mathbf{W}$ (dimensions $c \times (c+3)$) to process semantic and geometric data simultaneously. The input vector $\mathbf{x}_j = [\mathbf{f}_j, \Delta\mathbf{p}_j]$ leads to the output:
> $$
> \mathbf{y}_j = \mathbf{W} \mathbf{x}_j^\text{T}.
> $$
> Each output channel's contribution is calculated by the product of the weights and inputs, with Kaiming initialization setting $\mathbf{W}$ as a normal distribution $\mathcal{N}(0, \sqrt{\frac{2}{c+3}})$, limiting geometric data's influence due to its smaller proportional weight.
>
> In contrast, DSA segregates the handling of semantic and geometric data using two separate weight matrices, $\mathbf{W}_f$ for semantic (dimensions $c \times c$) and $\mathbf{W}_p$ for geometric data (dimensions $c \times 3$). This results in outputs:
> $$
> \mathbf{y}_j = \mathbf{W}_f \mathbf{f}_j + \mathbf{W}_p \Delta\mathbf{p}_j.
> $$
> The initialization $\mathbf{W}_f \sim \mathcal{N}(0, \sqrt{\frac{2}{c}})$ and $\mathbf{W}_p \sim \mathcal{N}(0, \sqrt{\frac{2}{3}})$ allows for a more balanced influence of geometric data, thus enhancing the network's ability to extract and utilize geometric information effectively.
>
> Following your feedback, we have included the additional explanations and comparative analyses in the revised version of the manuscript.

---

### Author Rebuttal · Authors · 2024-08-07

We thank all reviewers for their positive comments about the novelty (7sY3, zyd3), motivation (J8xV, zyd3), writing quality (7sY3, zyd3), and experiments (J8xV, 7sY3, aBR6, zyd3) of this work.

  As suggested by the four reviewers, we conduct sufficient additional experiments on our LinNet and demonstrate more inspirable abilities.

1. For the comments raised by reviewer **J8xV**, we elaborated on the core ideas and contributions of our model. We conducted experiments on the NuScenes dataset to explore a general model without explicit architectural choices, confirming that the model performs robustly in both configurations.
2. For the comments raised by reviewer **7sY3**, we highlighted the significance of our improvements. We added latency measurements to better demonstrate the model's efficiency and modified the figures based on constructive comments from the reviewer.
3. For the comments raised by reviewer **aBR6**, we conducted experiments to verify the necessity of Batch Normalization (BN) in the DSA module and addressed the complexity of the hash query.
4. For the comments raised by reviewer **zyd3**, we implemented cross-validation tests to assess the generalization capabilities of our model and measured its memory footprint.

In conclusion, to enhance the scalability of point-based methods, we developed Linear Net (LinNet). LinNet achieves more efficient local aggregation by leveraging spatial anisotropy and channel anisotropy separately. Additionally, by mapping 3D point clouds onto 1D space-filling curves, we performed downsampling and neighborhood queries with linearly reduced complexity.  LinNet **achieves dual improvements in both speed and accuracy, largely addressing the long-standing scalability challenges associated with point-based networks.** Extensive experimental results demonstrated the superiority of the methodology design, which also shows that **even without the support of any additional techniques (e.g., sparse convolution, attention, graph convolution), purely point-based methods can achieve good results by relying only on a simple MLP.** We anticipate that these insights will encourage the community to rethink methods for efficiently point clouds learning.

We include three figures in **the PDF** and cite them in rebuttal.

---

### Decision · Program_Chairs · 2024-09-25

**Decision:**

Accept (poster)

**Comment:**

This paper initially received borderline to positive ratings (BR, BA, WA, SA). The authors appreciated the novelty of the proposed point cloud feature learning approach but had some specific constraints regarding aspects of the method. Most of these concerns appeared resolved after the rebuttal, with 3 reviewers leaning towards acceptance (6,6,8). The only reviewer left on a BR was unresponsive during the rebuttal and discussion stage. Taken all this into account, the final recommendation is to accept this paper.